# A Pig Model to Assess Skin Lesions after Apomorphine Application

**DOI:** 10.3390/biomedicines11051244

**Published:** 2023-04-23

**Authors:** Vera Martin, Christian Knecht, Sophie Duerlinger, Barbara Richter, Andrea Ladinig

**Affiliations:** 1University Clinic for Swine, Department for Farm Animals and Veterinary Public Health, University of Veterinary Medicine, 1210 Vienna, Austria; 2Institute of Pathology and Forensic Veterinary Medicine, University of Veterinary Medicine, 1210 Vienna, Austria

**Keywords:** pig model, apomorphine application, skin lesions, dermatological model

## Abstract

Owing to their similarities, pigs are often used as experimental models for humans. In particular, the similarity of the skin allows them to be a good dermatological model. The aim of the study was to develop an animal model in conventional domestic pigs to evaluate skin lesions macroscopically and histologically after a continuous subcutaneous apomorphine application. A total of 16 pigs from two different age groups were injected with four different apomorphine formulations for 12 h daily over a period of 28 days into the subcutis, which was then evaluated macroscopically for nodules and erythema, as well as histologically. Differences in skin lesions between the formulations were found, with formulation 1 leading to the fewest nodules, least skin lesions, no lymph follicles, least necrosis, and best skin tolerance. Older pigs were easier to handle and, because of the thicker skin and subcutis of these animals, drug application with the appropriate needle length was safer. The experimental setup worked well and an animal model to assess skin lesions after a continuous subcutaneous application of drugs could be successfully established.

## 1. Introduction

Pigs, *sus scrofa domestica*, are among the most important species used as animal models for human research and are increasingly used for preclinical drug development studies. The most common pig models are established for the following diseases: diabetes mellitus, atherosclerosis, wound healing, and metabolic syndrome [1]. In addition, many dermal studies have been established and conducted in pigs. Although in most experimental dermatological studies, rats, mice, and rabbits are used, it has been shown that pig skin is most similar to human skin, even histologically [2]. Both the thickness of the different skin layers as well as the relative hairlessness and cutaneous blood supply are just a few examples demonstrating the similarity between the two species, although apocrine sweat glands are absent in pigs [3,4,5,6]. In addition, the pig has a fixed subcutaneous layer and dermal hair follicles like humans [7].

Pig skin also shows high similarity to human skin regarding wound healing, which happens primarily through re-epithelialization and not through wound contraction, as in other experimental animals; therefore, pigs are often used as percutaneous absorption, wound healing, and skin toxicology study models [8]. In general, the results of dermatological animal trials, especially in pigs, can be well transferred to humans.

However, conventional domestic pigs are described as difficult to handle owing to their size and body weight. In contrast to larger pig breeds, minipigs are often economically interesting because of their lower body weight at birth (350–450 g) and weight up to 35–40 kg when fully grown, which is advantageous for housing compared with domestic breeds, which can weigh up to 100 kg at the age of 4 months [6,9]. Minipigs are increasingly used for preclinical drug development as an alternative to larger pig breeds because of their smaller size and easier handling. In addition, minipigs such as the Göttingen miniature pig are raised under strictly controlled conditions and microbiological monitoring [10]. Some miniature breeds, such as the Göttingen miniature pigs, are dwarf breeds, which are comparable in proportions (e.g., bone length) to conventional domestic pigs. This is probably caused by a deficiency of insulin-like growth factor 1 (IGF-1) and is also known as “pituitary dwarfism” [11]. However, owing to the smaller size of the animals, there is also less skin available for skin studies and these types of minipigs can also be expensive to purchase.

Apomorphine is a non-selective D1 and D2 dopamine receptor agonist with an emetic effect that is used as a long-term therapy for late-stage Parkinson’s disease patients. Parkinson’s is one of the most important neurodegenerative diseases worldwide, which occurs as a result of a loss of dopamine-producing neurons in the brain. The major symptoms include bradykinesia, tremor, and rigidity. In addition to motor disorders, it can also lead to depression or psychosis, which severely limits the patient’s quality of life [12]. Parkinson’s is a slowly progressive and incurable disease. The gold standard for treating symptoms is levodopa. However, after a few years of therapy with levodopa, a decrease in the effectiveness of the drug is observed and, as a result, patients experience involuntary movements (*dyskinesias*), which can be reduced with additional medications such as apomorphine. Although there are several approved formulations that contain apomorphine, they all cause local skin reactions in almost all patients [13]. This fact may lead to termination of the apomorphine treatment, as described in a large Spanish study with 166 patients. In this study, four patients stopped the therapy owing to skin lesions such as nodules [14]. Furthermore, the appearance of nodules may result in decreased absorption of apomorphine, and the lesions may negatively affect patients psychologically because of their external appearance [15]. The cause of nodule formation remains poorly understood. However, it is assumed that there is a local inflammatory reaction of the subcutaneous fat tissue (*panniculitis*) [15]. Such nodules can additionally be painful and occasionally lead to infection with abscess formation, which then requires antimicrobial treatment or even surgical removal [13].

Several experiments on apomorphine-induced skin lesions and drug development have already been performed in minipigs [16,17], as well as one experiment that was performed in conventional pigs using magnetic-resonance-imaging-assisted evaluation [18]. The aim of the current study was to develop an animal model in conventional domestic pigs to evaluate skin lesions macroscopically and histologically after a continuous subcutaneous apomorphine application for 28 days. In addition, the current study compared the suitability of different age groups and different experimental setups. For the third experiment, older animals were used to minimize punctures and the application of the formulations into the muscles because punctures into the muscles were observed in the smaller animals. During the current study, four different apomorphine formulations were compared for their capability to cause skin lesions.

## 2. Materials and Methods

### 2.1. Animals and Study Design

In total, three trials with 16 female conventional pigs (*Sus scrofa domestica*) were performed according to Austrian animal welfare regulations. Ethics approval was granted for the studies by the institutional ethics and animal welfare committee and the national authority in accordance with sections §§ 26ff. of the Animal Experiments Act Austria, Tierversuchsgesetz 2012-TVG 2012. The experiments were authorized by the national authority under the numbers 68.205/0089-V/3b/2019, 68.205/0125-V/3b/2019, and BMBWF 2020-0.292.770.

Four and six animals at the age of eight weeks were included in the first and second trials (groups A and B), respectively, while the six animals used in the third trial were four months old (groups C and D). Prior to the start of the trial, all animals went through an adaptation and training phase with positive conditioning for 4 weeks (Figure 1). All animals came from the university-owned research and teaching farm, which is under constant veterinary control. The farm is well monitored, the PRRSV status is unsuspicious, and piglets are routinely vaccinated against PCV2 and *Mycoplasma hyopneumoniae* prior to the weaning at the age of 4 weeks. Only clinically healthy animals were selected for the study; during the study, all animals showed a physiological weight gain and did not develop any clinical symptoms. In all trials, the animals had permanent access to water (tap water) and were fed ad libitum with a commercial pig diet (FerkelKorn^®^, Garant, Pöchlarn, Austria). During the training phase, the animals were housed in pairs, while during the application phase, they were kept individually to prevent the protective suit/harness from being removed. To reduce social stress, the animals were constantly allowed to have direct nose to nose contact between the bars of the individual pens. The floor consisted of a solid, concrete floor with straw bedding as enrichment and nesting material. Each pig had a floor area of 4 m². No lighting program was used; however, the barn was lit by natural light (according to the Austrian Federal Act on the Protection of Animals) and additional light sources were used for 12 h per day. The stable temperature was measured daily. The average stable temperature in the first trial was 23 °C, while it was 17 °C in the second trial and 21 °C in the third trial. As the second trial was performed in October and, in this trial, as in the first experiment, smaller animals were used, additional heat lamps were provided for the piglets. After the animals arrived, they were randomly distributed among the pens. The complete experiment was performed blinded, i.e., the pump application system was filled with the formulations by an additional person. Therefore, the person assessing the macroscopic lesions was blinded to the treatment.

In order to attach the pumps for the subcutaneous injections close to the body of the animals, they were dressed in a bodysuit throughout each study day in the first and second trial. The pumps were kept in pockets on the back of the bodysuit (Figure 2). To improve the animal model for the apomorphine application, older and heavier pigs were included in the third trial to increase the thickness of the subcutaneous tissue. The bodysuit was replaced by a dog harness, on which the pumps were enclosed in pockets on the back (Figure 3).

One week prior to the start of each trial, piglets were anaesthetized by intravenous injection of ketamine hydrochloride (Narketan^®^, 10 mg/kg body weight, Vetoquinol Österreich GmbH, Vienna, Austria) and azaperone (Stresnil^®^, 1.3 mg/kg body weight, Elanco GmbH, Cuxhaven, Germany) for tattooing the application field to both sides of the neck. For each side of the neck, eight application fields, divided into four quadrants each, were tattooed, which determined the location of the application for the respective day. In addition, letters were tattooed to identify the application field (A–H on the left side of the neck and I–P on the right side of the neck, as shown in Figure 4 and Figure 5). A control field was added on each side of the neck, in which a needle without a catheter and application of a drug was placed for 12 h once a week (application field H on the left side of the neck and P on the right side of the neck). The size of each application field was 2.5 × 2.5 cm in the younger animals and 3 × 3 cm in the older animals. Prior to each pinprick, the respective application field was cleaned of dirt with compresses and disinfected with alcohol.

### 2.2. Application of the Formulation

Apomorphine, which is known to cause adverse effects such as skin nodules in humans [19,20], and which has been previously used in a minipig model to reproduce the same skin lesions [16], was selected as a reference for our studies. In addition, three new formulations of apomorphine were tested for their capability of provoking side effects like skin lesions and especially nodules.

In the first and second trial, group A received an apomorphine formulation (5 mg/mL) that is commercially available and group B received a higher concentrated (40 mg/mL), microemulsion-based apomorphine formulation. Two additional formulations were assessed in the third experiment. Group C received formulation 1 and group D received formulation 2 (Table 1; both aqueous formulations; formulation 2 had a lower organic excipient content than formulation 1). The apomorphine 40 mg/mL microemulsion formulation was tested to examine the microemulsion vehicle in comparison with the apomorphine 5 mg/mL. However, the first two trials showed that the microemulsion vehicle was not applicable as an excipient because it induced cell necrosis. Therefore, the following prodrug formulations were all aqueous formulations.

On the left side of the neck, all animals received the corresponding placebo formulations. This placebo formulation was identical to the apomorphine formulation, missing the active ingredient; the injection volume remained the same. During the application phase, all formulations were administered subcutaneously for 12 h per day for a period of 28 days. In addition, a needle, without a catheter and without application of a formulation, was placed in a control field on each side of the neck once a week for 12 h in order to see what macroscopical and histological changes were caused by a needle prick and the needle remaining in the skin for 12 h.

The formulations were applied via a pump application system, which is commercially available for Parkinson’s patients. This pump was attached to an infusion set with the needle for application at its end (Figure 6). While in the first trial, the needle was 8 mm long (Therastick 28G^®^, Fresenius KABI AG, Bad Homburg Germany), a shorter needle (Neria G27^®^, 6 mm, Unomedical A ConvaTec Company, Lejre, Denmark) was used in the second and third trials. The needle was fixed to the skin with tape (Animal Polster^®^, Snogg, Vennesla, Norway) in all three experiments to prevent premature loss of the needle (Figure 7). After the start of the application, the animals were checked every hour to detect premature loss of the needle.

As apomorphine has an emetic effect, an antiemetic drug (Motilium^®^ 10 mg film-coated tablets, JANSSEN-CILAG Pharma GmbH, 1020 Vienna, Austria) was administered perorally in a grape to the animals three times a day during the whole application phase.

### 2.3. Clinical and Terminal Observation

Throughout all studies, the inner body temperature of all animals was measured daily and the body weight was recorded once a week. Each application site was assessed twice daily, i.e., before and after the twelve-hour application, for nodules and erythema by visual examination and palpation (Table 2, Table 3 and Table 4). The scoring system was used from a previous trial with minipigs (data unpublished). For both nodules and erythema, the frequency of occurrence on the 28 injection days was assessed (groups A and B with five animals each—140 injection days per group and groups C and D with three animals each—84 injection days per group). If a nodule or erythema was felt or seen, the average recovery time in days was calculated. For presenting the results on nodules, a score was calculated for each animal by adding morning and evening observations for each side of the neck (Table 2). For results on erythema, the scores of both Table 3 and Table 4 were added for both observation time points (morning and evening) per animal and neck side. In addition, pictures were taken from each side of the neck in the morning and evening.

Starting in the second trial, the size of the nodules was additionally recorded in centimeters. 

At the end of each trial (study day 28), all animals were euthanized. For euthanasia, all animals were anaesthetised by intramuscular administration of ketamine hydrochloride (Narketan^®^, 20 mg/kg body weight, Vetoquinol Österreich GmbH, Vienna, Austria) and azaperone (Stresnil^®^, 1.3 mg/kg body weight, Elanco GmbH, Cuxhaven, Germany). Subsequently, the animals were euthanised by intracardial injection of T61^®^ (1 mL/10 kg, Intervet GesmbH, Vienna, Austria). The tattooed skin field was removed and fixed in 10% neutral buffered formaldehyde for one week. Several skin areas were trimmed (see numbers marked in red in Figure 4 and Figure 5), embedded in paraffin, sectioned, and stained with hematoxylin and eosin (HE) using standard protocols for histological assessment.

Selected injection time points were histologically examined for the four categories listed below (Table 5). For each category, the histological lesions were scored from 0 to 3 (0 = not present, 1 = mild, 2 = moderate, and 3 = severe). A sum of the individual scores was calculated for each histological section and for each animal. In addition, the sums of all sections were added up for a total histoscore per animal. The insertion depth of the needle was assessed for each histological injection site, and it was recorded if the pinprick ended in the subcutis, at the interface between subcutis and musculature, or if the needle went into the musculature. 

### 2.4. Statistical Analysis

Statistical analysis was performed with IBM SPSS^®^ Statistics (version 28.0.0.0, IBM, Armonk, New York, United States). The obtained data were tested for normal distribution using the Shapiro–Wilk test. When a normal distribution of the data was given, ANOVA with Bonferroni post hoc corrections was used to detect significant differences between the treatment groups (*p* < 0.05 was considered statistically significant). If the data were not normally distributed, non-parametric tests (Kruskal–Wallis test/Mann–Whitney U-test) were used to compare the results between the respective groups.

## 3. Results

### 3.1. Clinical Observation

No deaths or clinical signs of systemic toxicity were observed during the trials. Overall, there were no treatment-related differences in body weight, average daily weight gain, or feed intake between animals receiving the different formulations. During the application phase, no abnormalities in inner body temperature were observed in the animals and the temperature never exceeded 39.5 °C.

In the first two trials, the animals had an average weight of 18.73 kg (SEM: 0.75) at the start of the application phase and an average weight of 34.46 kg (SEM: 1.17) at the end of the trial. In the third trial, larger pigs with an average starting weight of 48.58 kg (SEM: 3.27) were included to prevent the needle from puncturing the muscle. On study day 28, these animals reached an average weight of 60.17 kg (SEM: 2.23). All pigs increased their body weight continuously. The average daily weight gain was 561.96 g/day (SEM: 7.95) for the animals in groups A and B and 413.69 g/day (SEM: 53.31) for the animals in groups C and D.

### 3.2. Macroscopic Parameters

#### 3.2.1. Nodules

Nodules occurred more frequently on the right side of the neck compared with the left side of the neck. Formulation 1, applied to pigs in group C, induced 11 nodules after 84 injections. In all other formulations, a nodule occurred after almost every injection day in almost every animal. Group A showed 127 nodules after 140 injections (see for e.g., Figure 8), group B had 129 nodules after 140 injections, and group D showed 83 nodules after 84 injections (see Figure 9). The average recovery time of nodules in group A was 3.54 days (SEM: 0.39) and that in group B was 2.86 days (SEM: 0.19). Group D showed nodules with an average recovery time of 5.90 days (SEM: 0.03). Animals from group C had an average recovery time of 0.64 days (SEM: 0.10). For the sum of nodules of the right side of the neck, all groups differed significantly (*p* < 0.001) from each other, except for groups A and B (*p* = 1).

Compared with the right side of the neck, the animals had fewer nodules after the placebo administration, which also remained for a shorter time. On average, the animals had 8 of 28 possible nodules (SEM: 1.60) and these nodules showed an average recovery time of 1.25 days (SEM: 0.19). The sum of nodules per day and per animal/group induced by the placebo is shown in Figure 10. Statistically, no significant differences between treatment groups were detected for the sum of nodules on the left side of the neck. The size of the nodules on the right side of the neck varied between the groups. In group A, the maximum size of nodules was between 1.5 and 2 cm and the nodules were seen for one observation time point. In comparison, all three animals in group B showed a maximum nodule size of 3 cm for one observation time point. Group C had very few nodules with a maximum size of 1 cm for one observation time point. Pigs in group D showed a maximum nodule size of 2–3 cm for 1–2 observation time points. The nodules became smaller over time and then disappeared. Similar results were obtained on the left side of the neck regarding the size of the nodules. In group A, the largest nodules were 1 cm; in group B, they were 2–3 cm; group C had only one nodule with 1 cm in diameter; and in group D, the maximum nodule size was 2 cm. 

#### 3.2.2. Erythema

Animals from group C showed no erythema on the right side of the neck. The five animals from group A showed an average of 11 erythema (SEM: 3.17, 55 erythema out of 140 possible erythema), which disappeared after an average of 1.27 days (SEM: 0.19). Group D showed an average of 10 erythema (SEM: 1.53, 30 erythema out of 84 possible erythema) with an average recovery time of 1.03 days (SEM: 0.17). Group B showed an average of 3.60 erythema (SEM: 1.13, 18 erythema out of 140 possible erythema) with an average recovery time of 0.85 days (SEM: 0.26). The summed erythema scores for each study day and each animal are presented in Figure 11. Statistically, no significant differences were detected between the different treatment groups.

On the left side of the neck, there was no erythema detected in any of the animals.

### 3.3. Histologic Assessment

Histological evaluation of samples of the first trial showed that 8 out of 64 punctures went into the interface between the subcutis and musculature and 14 out of 64 punctures went into the musculature. To prevent further punctures into the muscle, shorter needles (6 mm instead of 8 mm) were used for the next two trials. In the second experiment, 27 of 96 punctures reached the musculature and, in the third experiment, 10 of 96 punctures went into the muscle.

In some histologic slides, the needle prick itself (without any application of a formulation) and the disposition of the needle in the skin for 12 h in the control field caused a mild perivascular inflammatory reaction, a mild diffuse inflammatory reaction, and a mild necrosis of the fat tissue.

In comparison with the left side of the neck, higher histoscores were detected on the right side of the neck in most animals. Animal 7 (group A) had the highest total histoscore, with a sum of 78. In general, apomorphine 5 mg/mL, which is already commercially available, showed higher total histoscores than the other formulations. The formulation with the lowest total histoscores was formulation 1 (group C). Group C showed mild to moderate perivascular inflammation, absent to severe diffuse inflammation, no lymph follicles, and absent to extensive necrosis, with most injection sites showing no necrosis. The other formulations caused more frequent and more extensive necrosis (see Figure 12 for representative histologic images of lesions with varying severity). All groups showed a significant difference to group C (*p* < 0.001). There was no significant difference between the other treatment groups (*p* > 0.076–1).

On the left side of the neck, an average total histoscore of less than 30 was noted. Histologic lesions were mostly present as mild to moderate perivascular inflammation, none to moderate diffuse inflammation, and none to moderate necrosis. Mild lymph follicles were found at only one histologically examined injection time point on the left side of the neck. The total histoscore of the right and left side of the neck are presented in Figure 13 and Figure 14. For the total histoscore of the left side of the neck, groups A and B (*p* = 0.024), B and C (*p* < 0.001), and C and D (*p* = 0.024) differed significantly from each other. 

To obtain a better overview of the results of the different formulation a heatmap was created (see Figure 15).

## 4. Discussion

When parenteral administration of drugs is desired, injection/infusion therapy can result in adverse effects in up to 90% of cases [21]. The development of nodules is the most common adverse reaction following apomorphine injection and may affect up to 68% of all patients receiving such therapy [14]. Owing to the development of nodules, which has a negative effect on the human psychological state as a result of their external appearance, it is possible that the apomorphine therapy is discontinued [15,22]. As described in a previous study, nodules are caused by a local inflammatory reaction in the subcutaneous fat tissue [15]. In our study, more nodules were seen in animals that showed more lymph follicles and a higher histological score for diffuse inflammatory reaction and necrosis. The results of formulation 1, which led to the fewest nodules, least erythema, no lymph follicles, and least necrosis compared with the other formulations used in this study, confirmed better skin tolerance of this substance. However, the next step should be a pharmacokinetic evaluation of the formulations to detect the apomorphine concentration in plasma. The applied volume and all other substances of the placebo formulations led to the formation of single nodules, which were noticeable for a shorter time in comparison with the formulations with the active ingredient. Over the course of the different histologically assessed injection days, no notable differences in the histologic scores were found. 

As several trials have already been conducted using minipigs as experimental animals for continuous subcutaneous drug administration [16,17,23], the aim of the current study was to adapt this animal model to domestic pigs and prove that domestic pigs are also well suited for continuous subcutaneous infusion of drugs. 

A similar experiment with conventional pigs has already been performed, but in the course of this study, the apomorphine formulations were injected into the skin for only one day over 20 h, followed by both a magnetic resonance imaging and a histopathological examination of the skin [18]. The injection sites were examined shortly after the end of the injection and once a week (up to 4 weeks after the injection). In this study, a five-point scoring system for erythema and nodules was used for macroscopic changes. No macroscopic change could be seen or palpated, except for a small firm area (<1 cm), which was felt in the first 24 h after injection in 15–30% of the four treatment groups. In addition, an MRI was performed 2 and 4 weeks after the injection in order to be able to assess the course of the lesions. Histopathologic examination of the skin in the above-mentioned study was performed 4 weeks after the injection. However, comparable to our study, anything from minimal inflammatory reactions to severe necrosis was found in the subcutis. Additional techniques for visualizing the lesions and the recovery time of the lesions in vivo like magnetic resonance imaging should also be considered for further studies [18].

In the study by Ramot et al. 2018 [16], 16 Göttingen minipigs were included, weighing 18–20 kg, and a needle length of 8 mm was used. In this study, two different formulations (a commercially available formulation and a new formulation) were continuously applicated over 28 days. Moreover, blood was taken from the animals and the skin was examined histologically. In addition, selected organs (brain, heart, kidneys, liver, adrenal glands, and spleen) were examined microscopically. For the histologic assessment of the skin, a semi-quantitative scoring system (five-point scale) was used. Similar to our study, differences in the development of the nodules between the two formulations were found. Histologically, the formulations that caused more nodules showed increased inflammation and increased necrosis, as described in our study.

In our study, no pharmacokinetic evaluation or microscopic examinations of other organs were carried out, as the focus of our study was to replicate macroscopic and histological changes in the skin. The needle penetration depth was not evaluated histologically in any of the above-mentioned studies. Based on our experiences, it can be speculated that, in minipigs weighing 18–20 kg, a needle of 8 mm in length might have punctured the muscles.

As longer needles (8 mm) were used in the first experiment and needle puncture into muscle tissue occurred in 21.9% of the histologically evaluated needle pricks, shorter needles (6 mm) were used for the next experiments, both for the smaller animals in experiment 2 and for the older animals in experiment 3. In the second experiment, there were still several punctures into the muscle. In comparison, there were clearly fewer punctures into the muscle in the third experiment, but they were still present. The subcutaneous skin layer in the larger animals should be almost the same thickness as in an adult human and should be sufficient to allow application of the formulation into the subcutis. However, our suspicion is that the animals have pushed the needle deeper into the muscle by rubbing against the walls, resulting in limited small necrosis. During the hourly control, this behavior was never observed, but it cannot be excluded as the animals were unobserved for 20–30 min between the inspections. It is possible that some formulations were more unpleasant than others and caused an itch that could cause the animals to rub against the walls. Although a pinprick to the musculature caused necrosis by itself, the necrosis at the sites where one of the formulations was administered was much more extensive and severe than the necrosis from the pinprick alone. However, owing to limited necrosis, a pinprick into the muscle did not cause a nodule and could also be histologically distinguished from the severe histologic changes due to the application of the different formulations. It is always important to assess the depth of penetration of the needle, as administration of apomorphine is only recommended into the subcutis and not into the muscle. Owing to the fact that Parkinson’s patients have a lower weight at the beginning of the diagnosis than healthy persons of the same age group [24], and because they lose more and more weight over the course of the years [25], it cannot be excluded that the application of apomorphine does sometimes occur in the musculature in humans. In addition, in human medicine, the skin is rarely examined histologically for changes/nodules caused by the application of apomorphine.

In the current experiments, it was shown that both smaller and larger domestic pigs can function as an animal model for continuous subcutaneous infusion of drugs. However, the older animals were easier to handle after appropriate training and, because the skin of these animals was already thicker compared with younger animals, drug application with the appropriate needle length is safer. The tattoo with the specified application days worked very well and can be transferred to other animal models.

In several studies, a nigrostriatal toxin (N-methyl-4-phenyl-1,2,3,6-tetrahydropyridine), which acts as a precursor molecule of the mitochondrial toxin N-Methyl-4-phenylpyridine and destroys dopaminergic cells in the substantia nigra in the pig and human brain, has been shown to cause Parkinson’s-associated symptoms such as muscle stiffness, hypokinesia, and coordination disorders [26,27]. Our developed animal model could additionally be used in an experiment trying to reduce the induced symptoms. It would be interesting to combine such or similar animal models with the present model in domestic pigs because our model displays high potential for a wide range of areas in human medical research.

To the best of our knowledge, this is the first report of skin lesions after subcutaneous long-term infusion therapy with apomorphine in domestic pigs and may support the development of pig models for human diseases.

## Figures and Tables

**Figure 1 biomedicines-11-01244-f001:**
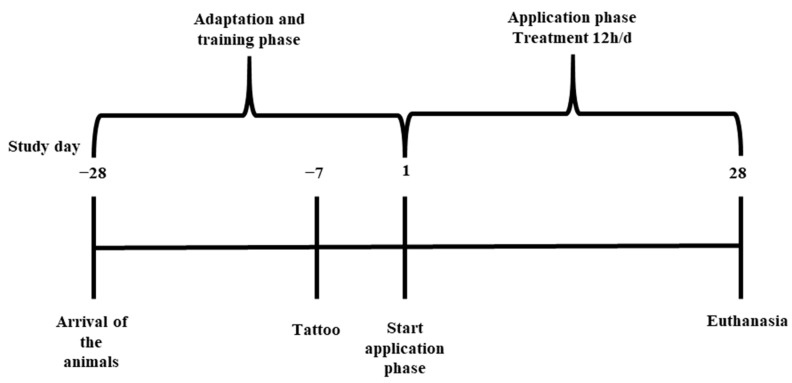
Timeline of the trials.

**Figure 2 biomedicines-11-01244-f002:**
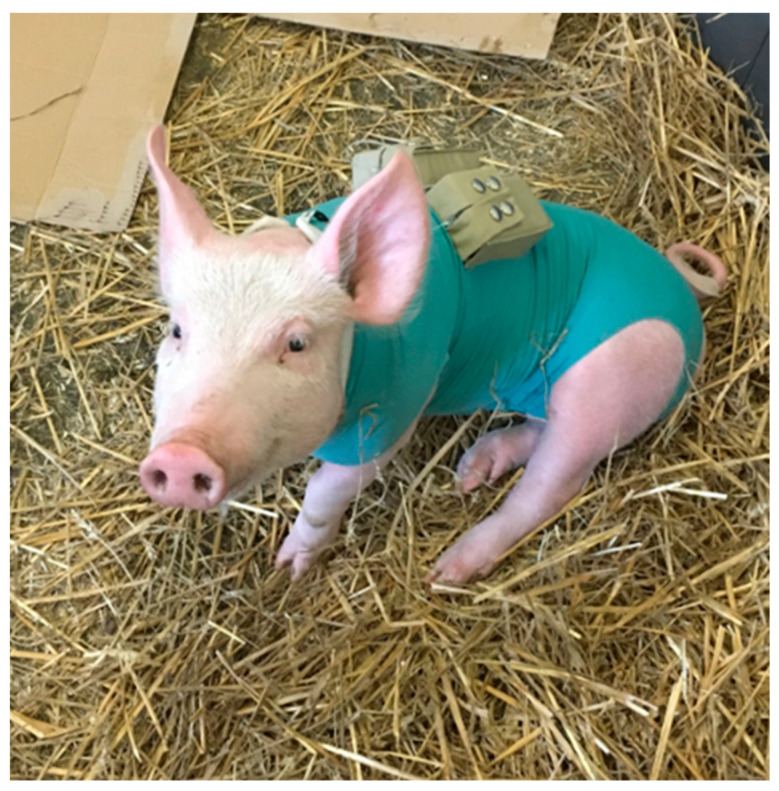
Animal with bodysuit and pumps in the pockets.

**Figure 3 biomedicines-11-01244-f003:**
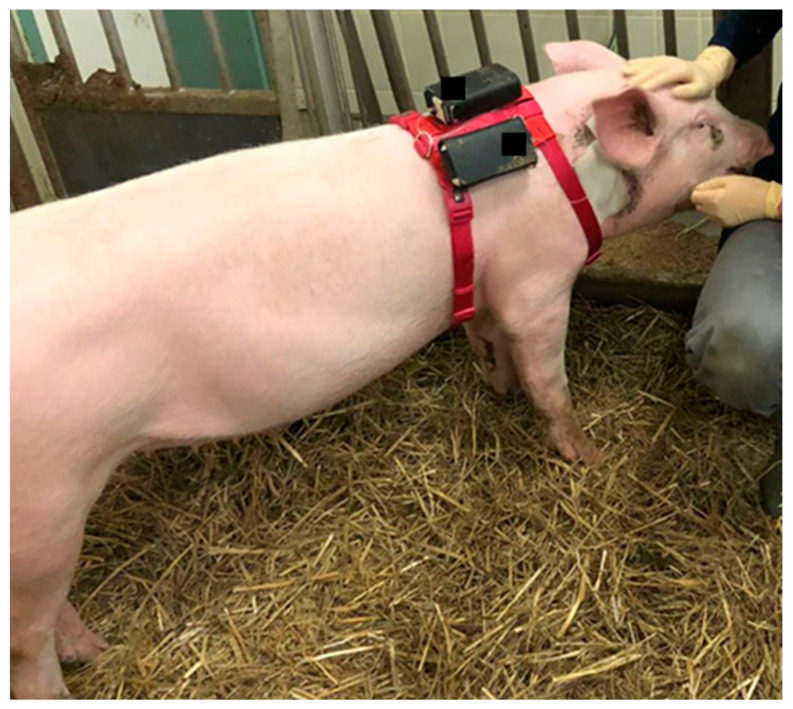
Animal with dog harness and pumps enclosed in the pockets.

**Figure 4 biomedicines-11-01244-f004:**
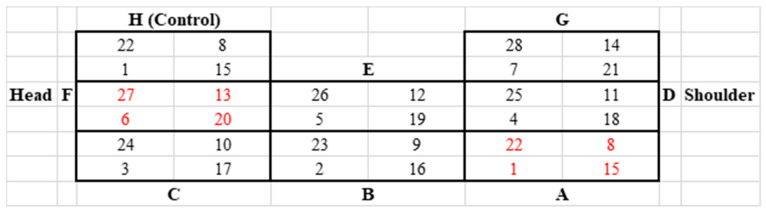
Tattooed field on the left side of the neck. Application fields A to H divided into four quadrants each. The numbers in the quadrants indicate the study day of application. Numbers in red indicate quadrants used for histological examination.

**Figure 5 biomedicines-11-01244-f005:**
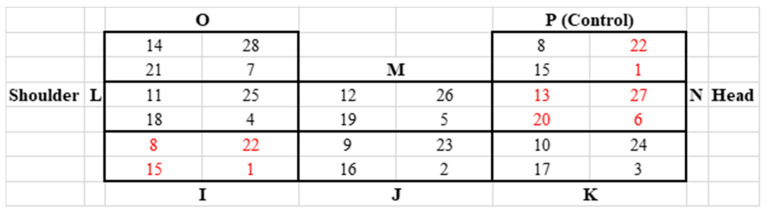
Tattooed field on the right side of the neck. Application fields I to P divided into four quadrants each. The numbers in the quadrants indicate the study day of application. Numbers in red indicate quadrants used for histological examination.

**Figure 6 biomedicines-11-01244-f006:**
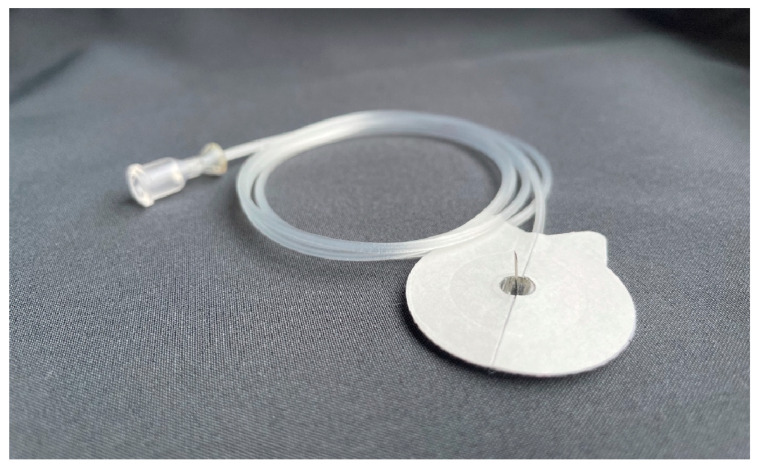
Infusion set with a needle for application at its end.

**Figure 7 biomedicines-11-01244-f007:**
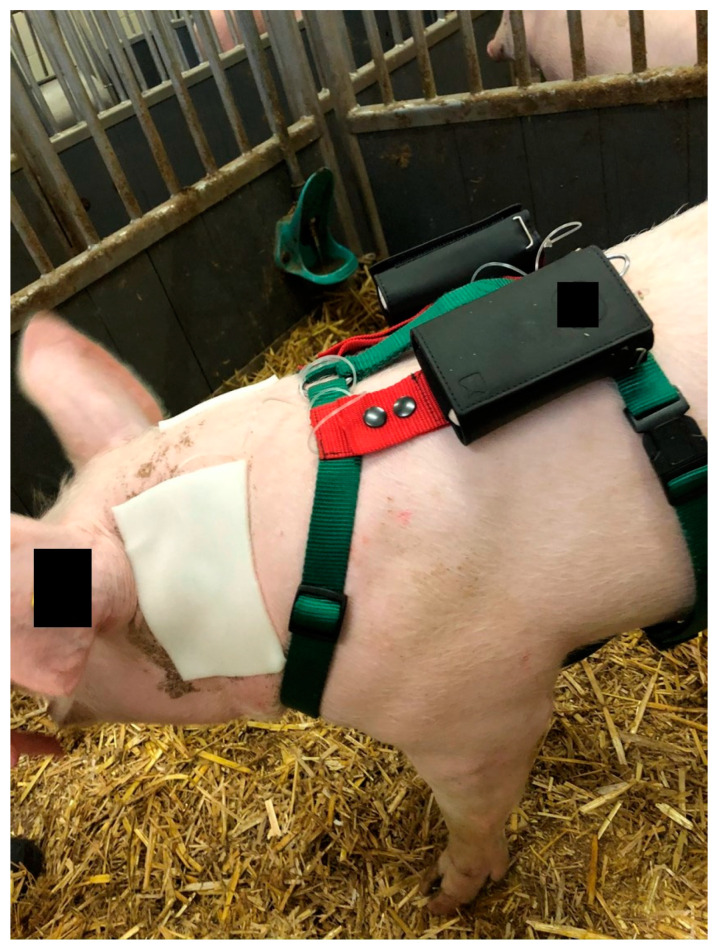
Fixation of the needle with Animal Polster^®^.

**Figure 8 biomedicines-11-01244-f008:**
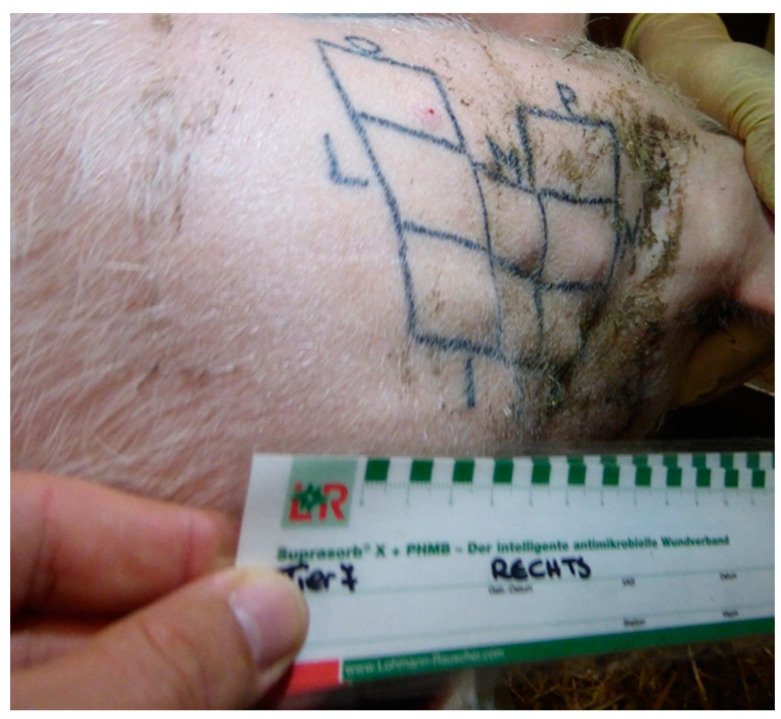
Animal 7 (group A, which received apomorphine 5 mg/mL) on study day 7 with multiple visible nodules on the right side of the neck.

**Figure 9 biomedicines-11-01244-f009:**
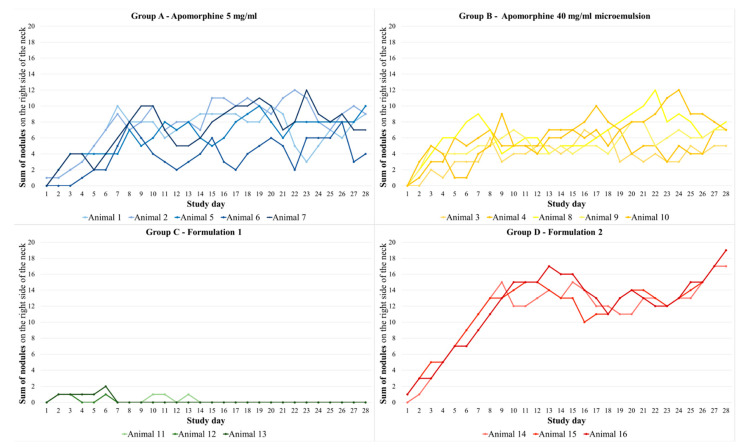
Course of the sum of nodules per study day and animal on the right side of the neck for all four treatment groups. The apomorphine formulations were administered to this side of the neck (sum of nodules results from the score described in Table 2 added for each side of the neck in the morning and in the evening). Group A—apomorphine 5 mg/mL; group B—apomorphine 40 mg/mL microemulsion; group C—formulation 1 7.13 mg/mL aqueous formulation; group D—formulation 2 7.54 mg/mL aqueous formulation. Statistically, all groups differed significantly from each other (*p* < 0.001), except for groups A and B (*p* = 1).

**Figure 10 biomedicines-11-01244-f010:**
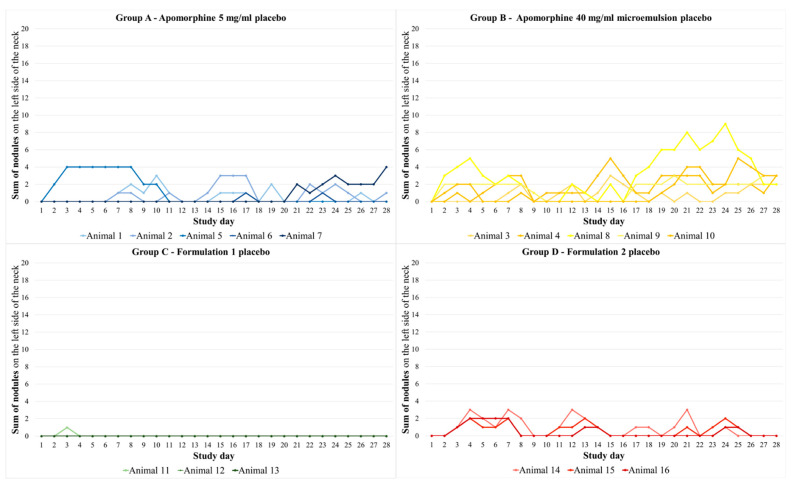
Course of the sum of nodules per study day and animal on the left side of the neck for all four placebo groups. The placebo formulations were administered to this side of the neck (sum of nodules results from the score described in Table 2 added for each side of the neck in the morning and in the evening). Group A—apomorphine 5 mg/mL placebo; group B—apomorphine 40 mg/mL microemulsion placebo; group C—formulation 1 7.13 mg/mL aqueous formulation placebo; group D—formulation 2 7.54 mg/mL aqueous formulation placebo. Statistically, no significant differences were detected between treatment groups.

**Figure 11 biomedicines-11-01244-f011:**
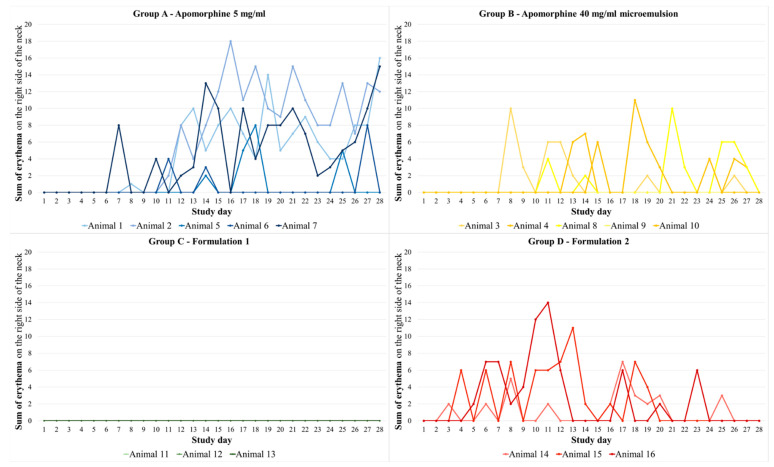
Course of the sum of erythema per study day and animal on the right side of the neck for all four treatment groups. The apomorphine formulations were administered to this side of the neck (sum of erythema results from the scores described in Table 3 and Table 4 added for each side of the neck evaluated in the morning and in the evening). Group A—apomorphine 5 mg/mL; group B—apomorphine 40 mg/mL microemulsion; group C—formulation 1 7.13 mg/mL aqueous formulation; group D—formulation 2 7.54 mg/mL aqueous formulation. Statistically, no significant differences were detected between the different treatment groups.

**Figure 12 biomedicines-11-01244-f012:**
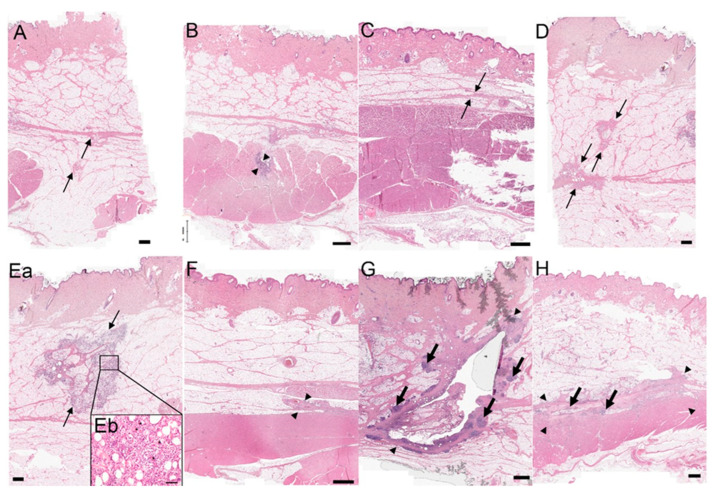
Representative histologic images of lesions with varying severity. All bars represent 1 mm (except image Eb bar = 80 µm). Image (**A**) (animal 11, study day 1): only the needle prick and the needle remaining (without injection of a formulation) in the skin for 12 h induced a mild perivascular inflammation (arrows) as a skin reaction to the foreign body/trauma. Image (**B**) (animal 11, study day 22): only the needle prick at the interface between subcutis and muscles and the needle remaining (without injection of a formulation) in the skin for 12 h induced a mild perivascular inflammatory reaction, a mild diffuse inflammatory reaction, and a small necrosis of the subcutis and the musculature (arrowheads). Image (**C**) (animal 8, study day 1): a moderate perivascular inflammatory reaction and a mild diffuse inflammatory reaction (arrows). Image (**D**) (animal 11, study day 13): a mild perivascular inflammatory reaction and a moderate diffuse inflammatory reaction (arrows). Image (**Ea**,**Eb**) (animal 11, study day 20): image (**Ea**) shows a mild perivascular inflammatory reaction and a severe diffuse inflammatory reaction (arrows); (**Eb**): higher magnification of the previous image; most cells are lymphocytes, plasma cells, and histiocytic multinucleated giant cells (asterisk). Image (**F**) (animal 5, study day 22): a moderate perivascular inflammatory reaction, a mild diffuse inflammatory reaction, and a small necrosis of the adipose tissue (arrowheads). Image (**G**) (animal 15, study day 6): a moderate perivascular inflammatory reaction, a moderate diffuse inflammatory reaction, severe lymphoid follicle formation (large arrows), and an extensive adipose tissue necrosis (arrowheads). Image (**H**) (animal 10, study day 7): severe histological lesions: a severe perivascular inflammatory reaction, a severe diffuse inflammatory reaction, moderate lymphoid follicle formation (large arrows), and an extensive necrosis of the adipose tissue and the musculature (area between arrowheads).

**Figure 13 biomedicines-11-01244-f013:**
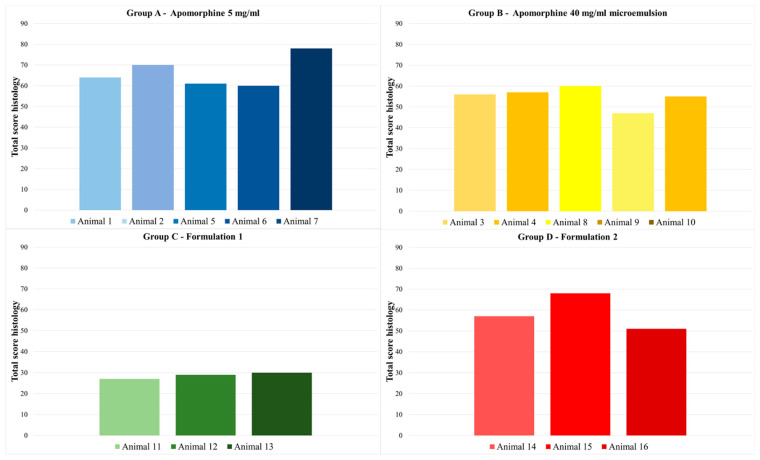
Total histoscore on the right side of the neck for all four treatment groups. The apomorphine formulations were administered to this side of the neck. Group A—apomorphine 5 mg/mL; group B—apomorphine 40 mg/mL microemulsion; group C—formulation 1 7.13 mg/mL aqueous formulation; group D—formulation 2 7.54 mg/mL aqueous formulation. All groups showed a significant difference to group C (*p* < 0.001). There were no significant differences between the remaining treatment groups (*p* > 0.076–1).

**Figure 14 biomedicines-11-01244-f014:**
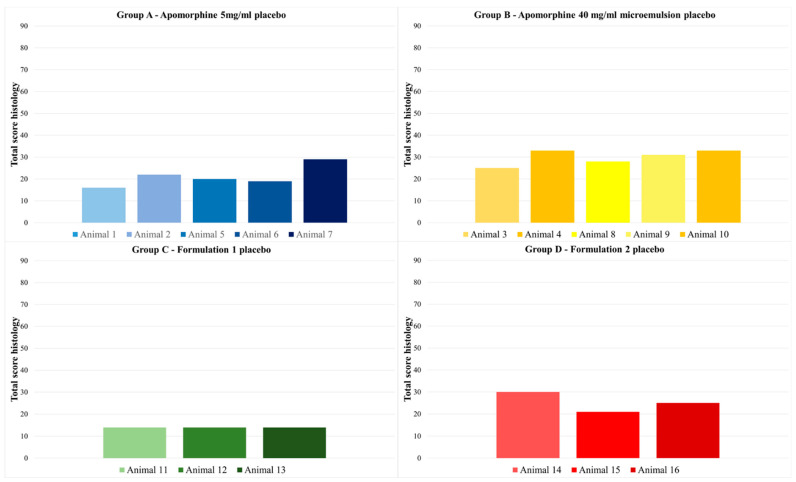
Total histoscore on the left side of the neck for all four placebo groups. The placebo formulations were administered to this side of the neck. Group A—apomorphine 5 mg/mL placebo; group B—apomorphine 40 mg/mL microemulsion placebo; group C—formulation 1 7.13 mg/mL aqueous formulation placebo; group D—formulation 2 7.54 mg/mL aqueous formulation placebo. For the total histoscore of the left side of the neck, groups A and B (*p* = 0.024), B and C (*p* < 0.001), and C and D (*p* = 0.024) differed significantly from each other.

**Figure 15 biomedicines-11-01244-f015:**
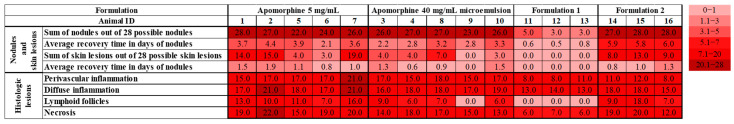
Heatmap to obtain an overview of the results of the different formulations. The heatmap shows both the macroscopic (sum of nodules out of 28 possible nodules, average recovery time of nodules in days, sum of erythema out of 28 possible erythema, and average recovery time of erythema in days) and histological (perivascular inflammation, diffuse inflammation, lymphoid follicles, and necrosis; each histologic lesion received a score from 0 to 3 (0 = not present, 1 = mild, 2 = moderate, and 3 = severe); the sum of all histologic scores was determined for the total histoscore per animal) results of the individual animals that received the four different formulations (group A—apomorphine 5 mg/mL; group B—apomorphine 40 mg/mL microemulsion; group C—formulation 1 7.13 mg/mL aqueous formulation; group D—formulation 2 7.54 mg/mL aqueous formulation).

**Table 1 biomedicines-11-01244-t001:** Assignment of the placebo and test formulations, dose, and volume administered to the respective animals/groups.

Group	Animal ID	Placebo Formulation Left Side of the Neck	Total Dose (mg) of Apomorphine per Day Left Side of the Neck	Test Formulation Right Side of the Neck	Total Dose (mg) of Apomorphine per Day Right Side of the Neck	Daily Volume (mL/12 h) Both Necksides
A	1, 2, 5, 6, 7	Apomorphine placebo	0 mg	Apomorphine 5 mg/mL	25 mg	5 mL/12 h
B	3, 4, 8, 9, 10	Apomorphine microemulsion placebo	0 mg	Apomorphine 40 mg/mL microemulsion	40 mg	1 mL/12 h
C	11, 12, 13	Formulation 1 placebo	0 mg	Formulation 1 7.13 mg/mL	45 mg	8.88 mL/12 h
D	14, 15, 16	Formulation 2 placebo	0 mg	Formulation 2 7.54 mg/mL	45 mg	8.88 mL/12 h

**Table 2 biomedicines-11-01244-t002:** Score for the number of nodules.

Score	Nodules
0	No nodules
1	1 nodule
2	2–5 nodules
3	6–10 nodules
4	>10 nodules

**Table 3 biomedicines-11-01244-t003:** Score for the degree of the severity of the erythema.

Score	Degree of Erythema
0	Not present
1	Minimal
2	Low
3	Moderate
4	High

**Table 4 biomedicines-11-01244-t004:** Score for the size of the erythema in percentage related to the affected application field.

Score	Size of the Erythema
0	No erythema visible
1	<10% of the field changed
2	10–25% of the field changed
3	26–50% of the field changed
4	51–75% of the field changed
5	76–100% of the field changed

**Table 5 biomedicines-11-01244-t005:** Score of the histological lesions.

Score	Histological Lesion
0, 1, 2, 3	Perivascular inflammation
0, 1, 2, 3	Diffuse inflammation
0, 1, 2, 3	Lymph follicles
0, 1, 2, 3	Necrosis

## Data Availability

All relevant data were included in the study. Raw data are available upon request by the authors.

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
