# Peer review of "A Pig Model to Assess Skin Lesions after Apomorphine Application"

_biomedicines, 2023, doi:10.3390/biomedicines11051244_

Round 1

Reviewer 1 Report

In the article “A pig model to assess skin lesions after apomorphine application”, Martin and coauthors aimed to develop an animal model in conventional domestic pigs to evaluate skin lesions macroscopically and histologically after a continuous subcutaneous apomorphine application. The importance of this research is undeniable. However, there are a number of minor remarks.

1) The volumes of injections differed by 9 times (1 mL for group B and 8,88 mL for groups C, D). A large volume of injected fluid may induce the development of macroscopic and histological effects. In my opinion, the investigated doses of drugs should be administered in the same volumes of the solvent.

2) Please, indicate in the materials and methods size of quadrants. If the size of the quadrants is small, a synergistic effect may occur when injected (especially large volume) into neighbor quadrants.

3) Please describe how the erythema grade was assessed.

4) The graph on the figures could be improved a bit. Contrast colors could be used for line of individual animals in each group.

5) Is there a correlation between punctures into the muscle tissue and the development of severe clinical and histological manifestations? This issue needs to be discussed.

6) In my opinion, a separate group of animals should be used for the administration of placebo. Since apomorphine is a lipophilic agent and can be distributed in the adipose tissue.

7) Figure 4 shows the both tattooed field for placebo and apomorphine.

8) The font in the figures 8, 9, 11, 12, 13 should be enlarged.

Author Response

  • The volumes of injections differed by 9 times (1 mL for group B and 8,88 mL for groups C, D). A large volume of injected fluid may induce the development of macroscopic and histological effects. In my opinion, the investigated doses of drugs should be administered in the same volumes of the solvent.

Thank you very much for this comment. There truly is a difference between the individual volumes of the formulations administered. These volumes had to be chosen to ensure a similar concentration of apomorphine. A larger volume could definitely increase the macroscopic and histological lesions. However, it was shown that despite the larger volume of formulation 1 or the placebos of groups C and D, the macroscopic and histologic lesions were fewer compared to all other formulations. So the applied volume did not influence the results.

  • Please, indicate in the materials and methods size of quadrants. If the size of the quadrants is small, a synergistic effect may occur when injected (especially large volume) into neighbor quadrants.

Thank you very much for this valuable comment. In order to avoid a synergistic effect, the quadrants were standardized and tattooed large enough with the help of a stencil. The size of the stencil was selected before tattooing the animals so that the area of each side of the neck was used to the maximum. Care was taken that the tattoo was not too close to the ear (as this part of the neck is very mobile) and ended in front of the shoulder blade. The exact size of the stencil was added in the Materials and Methods section (lines 126 - 127).

  • Please describe how the erythema grade was assessed.

Thank you very much for this comment. The degree of erythema was assessed subjectively, always by the same person, based on the severity of redness of the erythema. Minimal means that redness is evident but it is seen only with close observation. An Erythema was classified as low if mild redness was visible. Moderate means that significant redness was visible, but it was not as severe and fiery red as the erythema of the score high was assessed. For standardization, example pictures for each score were prepared during the first experiment, which were used all along the study to standardize the subjective evaluation of the severity of the redness.

  • The graph on the figures could be improved a bit. Contrast colors could be used for line of individual animals in each group.

Thank you for this comment. Indeed the images in the manuscript have a poor quality. This is due to the fact that the journal wants the images inserted in the manuscript before uploading. We would like to implement your suggestion to show the animals in each group with contrast colors. However, with the contrast color the group affiliation within the group would be lost. Therefore, we would like to remain with this representation of the figures. Of course, the images are additionally uploaded as tiff files which will be used in the final manuscript; so the quality of the final figures will be a lot better than the quality of the figures included in the word document.

  • Is there a correlation between punctures into the muscle tissue and the development of severe clinical and histological manifestations? This issue needs to be discussed.

Thank you very much for this question. This is indeed a very important point. There is a correlation between a prick in the muscle and the development of manifestations. We have shown that a pinprick to the musculature caused necrosis by itself, the necrosis at the sites where one of the formulations was administered was much more extensive and severe than the necrosis from the pinprick to the musculature alone (lines 400 - 402). However, a pinprick into the muscle does not yet cause a nodule due to limited necrosis and could also be histologically distinguished from the severe histologic changes due to formulation (implemented in the discussion, lines 402 - 405).

  • In my opinion, a separate group of animals should be used for the administration of placebo. Since apomorphine is a lipophilic agent and can be distributed in the adipose tissue.

Apomorphine is indeed a lipophilic agent, which is distributed in the adipose tissue. However, we only intended to evaluate the local skin reaction and not a systemic reaction in the animal. Another advantage was that there were no animal-specific differences, since each animal received both placebo and the formulation with the active ingredient, making it easier to compare the formulation with the placebo, without having animal-individual differences. In addition, we wanted to fulfill the 3Rs of animal experiments and therefore we only chose such a small number of animals and did not choose an extra group of animals for the placebos.

  • Figure 4 shows the both tattooed field for placebo and apomorphine.

Thank you for the remark. We must have made a mistake during the uploading process. The additional graphic was deleted.

  • The font in the figures 8, 9, 11, 12, 13 should be enlarged.

Thank you, the suggestions have been implemented.

Reviewer 2 Report

The manuscript by Vera Martin et al. introduced a model to assess skin lesions after apomorphine application by conventional domestic pigs. They found a formulation of the drug which could reduce all the side effects compared to others. Here I have several concerns for this study.

1. Any formulation of drug given to the pig is to mimic the use of the drug in human patients, so it would make much more sense to detect the drug concentrations of different formulations.

2. Data of nodules and erythema as well as histoscores were recorded very well at different time points from all of the pigs of different groups. Why don't you do statistics analysis with the data of Fig8 - Fig13?

3. The scores of Table5 should be 0,1,2,3.

4. The table between line128 and line 129 should be deleted.

Author Response

  • Any formulation of drug given to the pig is to mimic the use of the drug in human patients, so it would make much more sense to detect the drug concentrations of different formulations.

Thank you very much for this comment. The apomorphine concentration of the formulations was added in Table 1 (lines 149 - 150). Indeed, it would have been very useful to determine the drug concentration in the animal via blood. Unfortunately, however, this was not performed because we were primarily focused on creating an animal model to assess skin lesions, therefore no investigations on drug concentrations in the animals were made.

  • Data of nodules and erythema as well as histoscores were recorded very well at different time points from all of the pigs of different groups. Why don't you do statistics analysis with the data of Fig8 - Fig13?

Thank you for this comment. Statistical analyses on data of fig. 8 – 13 were performed and results included in the manuscript.

  • The scores of Table5 should be 0,1,2,3.

Thank you very much for this comment. The scores of table 5 has been changed accordingly (lines 202-203).

  • The table between line128 and line 129 should be deleted.

Thank you for the remark. We must have made a mistake during the uploading process. The additional graphic was deleted.
